# *Prevent with Pleasure*: A systematic review of HIV public communication campaigns incorporating a pleasure-based approach

Luke Muschialli[1]*, Jessie V. Ford[2], Lianne Gonsalves[3], Robert Pralat[4]

1 Department of Public Health and Primary Care, University of Cambridge, United Kingdom, 2 Department of Sociomedical Sciences, Mailman School of Public Health, Columbia University, New York, United States of America, 3 Department of Sexual and Reproductive Health and Research, UNDP-UNFPA-UNICEF-WHO-World Bank Special Programme of Research, Development and Research Training in Human Reproduction (HRP), World Health Organization, Geneva, Switzerland, 4 Department of Public Health and Primary Care, THIS Institute, University of Cambridge, United Kingdom

* Lgm35@cam.ac.uk

## Abstract

Previous research suggests integrating pleasure into HIV prevention programming improves health outcomes. There are no existing reviews on how exactly pleasure is used within HIV public communications campaigns (PCCs). This manuscript investigates: (1) how HIV PCCs operationalise pleasure; and (2) the efficacy of pleasure-based HIV PCCs. EMBASE, Web of Science Core Collection and PsycINFO were searched for articles that present pleasure-based HIV PCCs on 13/12/2023 (PROSPERO ID: CRD42023487275) with no language restrictions. A narrative synthesis on pleasure operationalisation centred around three inductively coded categories: *Enjoyment*, *Emotional Connection* and *Empowerment*. Another narrative synthesis summarised efficacy data around six categories of HIV-related outcomes. 19,238 articles were retrieved, with 47 articles included in analysis, describing 29 campaigns. 65.5% of interventions operationalised *Empowerment*, 48.3% *Enjoyment*, and 31.0% *Emotional Connection*, with narrative synthesis highlighting the diverse ways this was achieved across target communities. An analysis of efficacy identified heterogeneous outcome reporting with inconsistent results across studies, but important outcomes, such as stigma reduction and condom use, were positively associated with intervention exposure across all relevant interventions. We highlight a range of mechanisms through which pleasure can be operationalised, which should inform future intervention development, even if the extant literature weakly supports the efficacy of such interventions.

## 1. Introduction

HIV continues to be of paramount importance for global health. Despite the number of HIV infections having reduced by 60% since the peak of the epidemic in 1995, 1.3 million people were newly infected in 2023 [1], with infection disproportionately affecting key populations such as transgender ('Trans') women and men who have sex with men (MSM) [1]. Although global incidence has been decreasing across the past decade, it is now stagnating

**Data availability statement:** All data is available within the provided manuscript.

**Funding:** The authors received no specific funding for this work.

**Competing interests:** The authors have declared that no competing interests exist.

in many settings, and in some areas, such as regions of central Asia, it is even rising at alarming rates [2]. We are moving into an important era of HIV's history, with the Joint United Nations Programme on HIV/AIDS (UNAIDS) setting the ambitious target of ending the AIDS epidemic by 2030 [2]. Current prevention programming, however, is not on track to achieve this, with insufficient access to and uptake of prevention, such as pre-exposure prophylaxis (PrEP) [3–5]. As we move into the next, and potentially final, era of the HIV's history, it is vital that we are utilising the most efficacious prevention mechanisms to reach these ambitious goals.

Public communication campaigns (PCCs) are a public health tool with important implications for global HIV prevention. PCCs can be broadly defined as: (1) purposive attempts, (2) to inform or motivate behaviour change, (3) in a well-defined audience, (4) for non-commercial benefits, (5) within a given period, (6) by means of organised communication activities [6,7]. PCCs have a range of benefits for global HIV prevention programming, including reaching a wide range of audiences with standardised messaging, or targeting specific communities with refined messaging through multi-media platforms [8]. PCCs are also low cost-per-participant; an important considerations for the upscaling of prevention in low- and middle-income settings [9]. A 2014 meta-analysis found exposure to HIV mass media campaigns was associated with increased condom use and HIV transmission knowledge, particularly for countries that scored lower on the Human Development Index [8] and a 2006 meta-analysis conducted on HIV PCCs in 'developing' countries also found positive associations between intervention exposure and condom use [10].

There is no universally accepted definition of sexual pleasure [11]. Historically, discussions around sexual pleasure have focused on physical sensations, including nervous and hormonal responses [12]. More recent conceptualisations have begun to integrate emotional and cognitive facets of pleasure [13], however, these definitions have largely negated the importance of sexual rights [14,15]. The World Health Organisation (WHO) define sexual rights as the set of human rights that must be respected, protected and fulfilled to achieve sexual health, including the rights to equality and non-discrimination [16]. The experience of pleasure is dependent upon on the existence of sexual rights that allow for heterogeneity and autonomy in sexual experience [17–20]. Throughout this manuscript, therefore, the following working definition of sexual pleasure will be used, derived from existing working definitions from medical governing bodies [20–24]:

Sexual pleasure describes the physical, emotional and/or psychological satisfaction derived from erotic experiences and/or emotional connection, grounded in experiences of self-determination, safety, trust, non-discrimination, autonomy, privacy, communication and ability to negotiate sexual relations.

The term *'pleasure-based'* will be used to describe interventions that operationalise this definition.

Sexual pleasure has long been overlooked within sexual and reproductive health and rights (SRHR) [11,20,25]. A 2018 Guttmacher-Lancet commission on SRHR identified that sexual pleasure was 'largely absent from organised SRHR programmes and [its] links to reproductive health unstudied' [20]. Historic medicalisation and stigmatising of sex has led to the predominance of a risk-focused framing of sexual behaviour within contemporary HIV prevention programming, which often only highlights the risks of behaviours associated with HIV transmission, assuming that sexual decision-making is driven solely by rational health considerations [14], negating considerations of pleasure, connection and intimacy [4,26–28]. Despite risk having an important role in informing sexual behaviour, risk-focused programming does not adequately capture the importance of sexual pleasure in driving sexual well-being [29–31] and decision-making [32,33]. Sexual pleasure drives engagement with HIV protective

behaviours, such as PrEP use [33], as well as contributing to disengagement from prevention. For example, concerns about compromised pleasure have been noted to contribute to reduced condom use [34,35]. Without adequately incorporating pleasure into prevention programming, public health risks perpetuating a mismatch between sexual decision-making and intervention messaging, resulting in interventions with potentially limited sustained health behaviour change [30,31].

Studies investigating pleasure-based interventions have found consistently improved attitudes and knowledge about sexual health, partner communication and condom use among those exposed to interventions [36–41]. A 2006 meta-analysis found interventions promoting condom use that integrated condom eroticisation were associated with statistically significant improvements in condom use, HIV-related knowledge and partner communication [15]. A 2022 systematic review and meta-analysis also found pleasure-based interventions promoting condom use were associated with positive, statistically significant increases in condom use [14]. With strong theoretical and empirical evidence supporting pleasure-based interventions, public health researchers [31,42–46] and governing bodies, including the WHO [47], have endorsed the importance of sexual pleasure and encouraged the explicit inclusion of pleasure within SRHR prevention.

Increasingly, there has been a call to upscale HIV PCCs that operationalise pleasure [4]. *#PrEP4Love*, a Chicago-based PCC promoting PrEP, is centred around a sex-positive framing of HIV prevention, presenting images of couples of diverse racial, sexual and gender identities embracing, coupled with epidemiological language of disease transmission that have been re-appropriated to promote sex-positive messaging (e.g., *"transmit love"*) [48]. The intervention has been associated with statistically significantly increased engagement with discussions around HIV and uptake of PrEP [49,50].

There are, however, notable challenges to the implementation of pleasure-based HIV PCCs, primarily due to an underestimation of the efficacy of such interventions from the public and policy-makers [51], opposition to the financing of interventions that openly affirm sexual pleasure [31,42,52], and a heterogeneous, dissipated extant literature base which is insufficient to petition policymakers on the efficacy of such interventions when advocating for their upscaling. No previous evidence syntheses have been conducted on the efficacy of pleasure-based HIV PCCs, nor on how such interventions are defining and operationalising pleasure. Understanding the efficacy of interventions, and how most effectively to operationalise pleasure across different settings, intervention modalities and target populations, is necessary for the upscaling of pleasure-based interventions and accessing the aforementioned benefits of pleasure-based medicine.

This objectives of this manuscript are to:

1. Investigate how HIV PCCs operationalise pleasure, identifying common themes across interventions and exploring how this varies across target communities and settings, and;

2. Evaluate the efficacy of pleasure-based HIV PCCs, through synthesising quantitative and qualitative data across a range of behavioural (i.e., partnership behaviour), biomedical (i.e., PrEP use) and attitudinal (i.e., stigma) outcomes relating to HIV/AIDS.

## 2. Materials and methods

### 2.1. Search strategy

This review is compliant with PRISMA reporting guidelines [53] and registered with PROSPERO (ID: CRD42023487275). S1 File reports the PRISMA checklist.

Web of Science Core Collection, EMBASE and PsycInfo were searched from 01/01/2010-13/12/2023. This start date coincides with the UNAIDS announcement of their goal of achieving zero HIV discrimination and zero AIDS-related deaths [54]. Title/abstract restrictions were placed on all search concepts, with no language restrictions.

Search terms were derived from previous systematic reviews on adjacent topics [8,14,15]. Search terms related to two concepts: HIV/AIDS and PCCs (Table 1). No search terms relating to pleasure were included in the search due to the risk of excluding interventions that were not indexed for pleasure, did not self-identify as pleasure-based, or did not include terms relating to pleasure in the title/abstract. An assessment of whether interventions operationalised pleasure was therefore completed at full-text screening, matching the recommendations of previous reviews [11,14]. S1 Table presents the full search strategy.

## 2.2. Secondary search strategy

Single title/abstract and full-text screening was conducted, as opposed to gold-standard dual-screening. To reduce the risk of falsely excluding relevant studies due to an increased vulnerability to human error [55], the following comprehensive secondary search strategy was implemented [56]:

1. **Contacting field experts**. Researchers in sexual pleasure, HIV/AIDS and PCCs were contacted with information about article inclusion criteria, asking for recommendations.

2. **Hand-searching journals.** Journals which published articles included after full-text screening were extracted. The five most common journals were hand-searched across the period of interest, with relevant articles undergoing screening.

3. **Backwards and forwards citation chasing**. The software *Spidercite* [57] automated backwards and forwards citation chasing on all studies included after full text screening and relevant articles underwent screening.

4. **Repeat search using included campaign names**. The campaign names of included interventions were re-entered into the aforementioned search engines and identified papers were screened.

5. **Identifying relevant campaign names from *Pleasure Project's Pleasure Map* [58]**. *Pleasure Project* is an international advocacy organisation advocating for the eroticisation of safer sex [24]. They have developed a database of pleasure-based campaigns, *The Pleasure*

Table 1. Search concepts and terms.

| Search Concepts | Search Terms |
| --- | --- |
| *Theme 1*: HIV/AIDS | "human immunodeficiency virus" OR "HIV" OR "acquired immune deficiency syndrome" OR "AIDS" OR "sex practice*" OR "condom*" OR "PrEP" OR "Pre-exposure Prophylaxis" OR "sexual behaviour*" OR "safe* sex" OR "circumcision" |
| *Theme 2*: Public Communication Campaigns | "mass media" OR "television" OR "radio" OR "cinema" OR "movie*" OR "social media" OR "social network*" OR "publicity campaign" OR "campaign*" OR "public communication" OR "public service announcement" OR "newspaper" OR "magazine" OR "entertainment education" OR "social market*" OR "brochure" OR "flyer" OR "educational literature" OR "billboard" OR "Twitter" OR "Facebook" OR "Instagram" OR "YouTube" OR "Snapchat" OR "blog" OR "mobile app*" OR "text messag*" OR "sex* education*" OR "reproducti* education" OR "comprehensive sex* education" |

*Map* [58], which was searched for relevant HIV PCCs. Relevant campaign names were extracted and entered into the aforementioned search engines, and identified articles were screened.

6. **Extraction of studies from relevant reviews**. Although evidence syntheses were not included in analyses, reviews were flagged, and papers included in these reviews were screened.

## 2.3. Screening process, inclusion and exclusion criteria

Automated de-duplication of studies was conducted in the systematic review management software, *Covidence* [59]. The systematic review management software, *Rayyan* [60], was used to manage all subsequent screening.

At title/abstract screening, we included studies that: (1) were published in a peer-reviewed journal, (2) presented or evaluated HIV PCCs. A rapid full-text screen then assessed pleasure operationalisation, in which the sections of manuscripts addressing intervention programming were screened. A final full-text screen confirmed whether articles met all inclusion criteria.

Studies were excluded if they:

- **Investigated aspects of SRHR not related to HIV**. Studies investigating SRHR unrelated to HIV, such as sexually transmitted infections (STIs), were excluded. Papers that presented or evaluated SRHR campaigns, but addressed prevention relevant to HIV, such as condom use, were included.

- **Focused on non-sexual routes of transmission**. Studies investigating non-sexual HIV transmission, such as injection drug use, were excluded, due to the focus of this study on *sexual* pleasure.

- **Addressed general PCC exposure**. Studies relating general exposure to PCCs rather than assessing the influence of one specific campaign were excluded, due to an inability to desegregate the impact of pleasure-based PCCs.

- **Were systematic reviews, abstracts or conference proceedings**.

For title/abstract screening, all non-English language studies provided both titles and abstracts in English. No studies that did not provide an English full-text manuscript met the inclusion criteria for full-text screening.

## 2.4. Data extraction

Data extraction was carried out using a pilot data extraction table in Excel created by LM, collecting the following data:

- **Bibliometric information**; first author, year of publication, country of authorship team.

- **Study and intervention information**; population demographics, study design, intervention description, country, duration, target population, pleasure operationalisation, details of pre-testing.

- **Intervention efficacy**; outcomes measures assessed, details of outcomes.

Studies' risk of bias was assessed using one of the following tools: Joanna Briggs Institute's Instruments [61], the Mixed Methods Appraisal Tool [62] and the National Institute of Health [63] (see S2 Table).

For studies evaluating non-English language campaigns, campaign messages were translated using the automated translation software, *DeepL* [64]. This was done in consultation with native speaking colleagues.

## 2.5. Data analysis

A narrative synthesis [65] summarised pleasure operationalisation in included campaigns. During data extraction, three codes were inductively derived describing emerging common themes concerning pleasure operationalisation:

- **Enjoyment;** describing the pleasure associated with sexual activity, including the anticipation of physical pleasure, desire, and seduction.

- **Emotional Connection;** describing love, trust, reciprocity, and affection, both generally and in reference to partners.

- **Empowerment;** describing the experience and affirmation of sexual rights, autonomy, de-stigmatisation, and equality.

Throughout data extraction it was determined that outcome reporting was too heterogeneous for a meta-analysis to be performed. Therefore, a narrative synthesis was conducted on outcome data from studies evaluating intervention efficacy. Results were organised around the following inductively derived codes of HIV-related outcomes:

- **Condom Use;** including condom use with a new partner, number of condomless sex acts or condom use over a defined time period.

- **Biomedical Prevention;** summarising engagement with, intention to take and adherence to primary (i.e., PrEP), secondary (i.e., post-exposure prophylaxis (PEP)) and tertiary (i.e., antiretroviral therapy (ART)) biomedical prevention.

- **HIV Testing;** addressing both testing engagement and intention to engage with either self-administered or clinic-based testing.

- **Partnership Behaviour;** including new partnerships, number of partners and engagement in transactional sex.

- **Attitudinal Change;** including changes in stigma, self-efficacy and knowledge relating to HIV/AIDS.

- **Health Behaviour Change;** addressing outcomes related to HIV not captured by other codes, including voluntary medical male circumcision (VMMC), abstinence and sexualised drug use.

References to statistical significance refer to significance at the 5% level.

## 3. Results

## 3.1. Study characteristics

Our search identified 19,238 records, and following de-duplication, 11,695 records underwent title/abstract screening. 939 articles underwent full-text screen, with 35 papers meeting inclusion criteria. A further 12 articles were retrieved through the secondary search. The final sample included 47 articles, describing 29 campaigns (Fig 1).

14 included articles were RCTs (29.8%), 10 cross-sectional (21.3%), 8 qualitative (17.0%), 6 quasi-experimental, 6 pre-post studies (12.8%, respectively), and 3 mixed method (6.9%). Interventions were based in United States of America (USA, n=15, 51.7%), China (n=3,

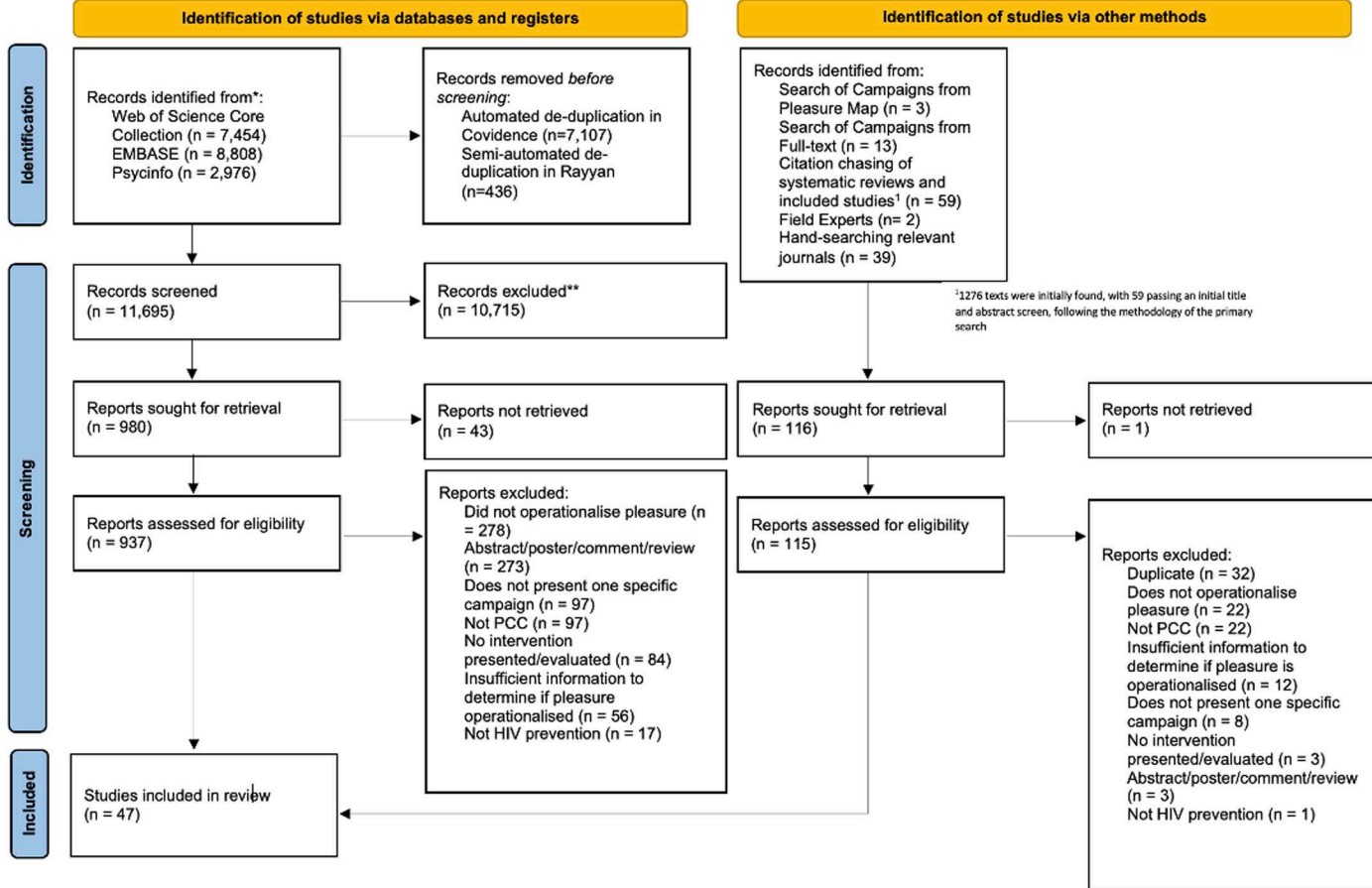

**Fig 1. PRISMA Flow Diagram describing the primary and secondary screening for the review.**

10.3%), Ethiopia, Scotland, the United Kingdom (UK), Zambia, India, Sweden, Mexico, Australia, Italy, Singapore and Uganda (n=1, 3.4%, respectively). Targeted campaigns focused primarily on young people (n=15, 51.7%), MSM (n=11, 37.9%), people of colour (n=6, 20.7%), cisgender women (n=5, 17.2%) and Trans women (n=4, 13.8%). S3 Table presents key features of included studies and operationalisations of pleasure, while S4 Table presents key outcomes of included studies. S2 Table reports quality assessment. 53.2% of studies had a low risk of bias (n=25) and 46.8% had a moderate risk (n=22). No studies had a high risk of bias. S5 Table presents the studies excluded from the primary full-text screening, with reasons for exclusion.

### 3.2. Operationalisations of pleasure

**3.2.1. Empowerment.** 19 interventions operationalised empowerment. These interventions focused on young people (n = 12; 63.2%), MSM (n = 4; 21.1%), cisgender women (n = 4; 21.1%) and Trans women (n = 2; 10.5%) and were based in USA (n = 10; 52.6%), Uganda, Ethiopia, Singapore, Italy, Mexico, Sweden, India, and UK (n = 1; 5.3%, respectively).

Ten interventions empowered individuals to engage with protective behaviours more effectively and to negotiate prevention, specifically condom use. Interventions such as *Safetxt*

[66,67] and *healthempowerment.org* [68] advocated for self-efficacy in negotiating preventive behaviours and communicating prevention preferences. *MyPEEPS Mobile* [69–74], *Project-HeartforGirls.com* [75] and *Skyddalaget* [76] provide specific strategies for negotiating condom use. For example, *ProjectHeartforGirls.com* use an interactive exercise that involves direct instruction, role-playing and assessment to build skills in sexual assertiveness [75]. These principles are also enacted in HIV testing interventions. *Texting 4 Sexual Health* promotes self-efficacy for HIV testing, with one message instructing participants to *'Take control of your sex life. Ask your doctor to be tested for ST[I]s and HIV'* [77].

Interventions also emphasise community benefits of prevention. The untargeted, Italian national prevention campaign, *United Against AIDS*, focuses on collective responsibility to end AIDS, with taglines such as *"AIDS is everyone's concern"* and *"We are all in this together"* [78]. *Testing Makes Us Stronger* use messages such as *'Our HIV status is powerful information'* and *'Gay and bisexual men are standing up against HIV'*, employing collective pronouns to promote togetherness and community mobilisation [79–81]. Community mobilisation was also achieved through integrating relevant cultural phenomena in interventions. *LifeSkills Mobile*, an app targeting young Trans women, created a component called the *House of LifeSkills*, which incorporated urban house and ball culture into prevention messaging, channelling the intrinsically community-centred aspects of ball culture [82].

Some interventions integrated the importance of sexual rights, with *Nalamanda's Radio and Theatre Programme* centring messages around women's rights in the context of HIV [83]. *Just/US*, a Facebook intervention targeted at young people, was based around the core principle that sexual health is a human right and a function of social justice [84].

Interventions also de-stigmatised prevention. *Oral HIV/AIDS Prevention Messaging for illiterate Ethiopian Women* included discussions between women on the importance of safe sex, highlighting common myths and stigma around condom use, including messages such as *'I feel shy to carry a condom. People may be suspicious of me doing adultery'*. This is then counteracted, affirming that buying condoms *'is a sign of being wise and intelligent'* and key to accomplishing your *'plan and desire'* [85]. *Safetxt* similarly employs a non-judgemental, non-stigmatising approach to sharing information about STIs [66,67].

Interventions notably demonstrate an integration of elements of risk in a sex-positive manner. For example, a poem in *Oral HIV/AIDS Prevention Messaging for illiterate Ethiopian Women* acknowledged that *'You feel proud to enjoy sex'* but followed this was a reminder that *'You have to know that you expose yourself to HIV/AIDS'* [85]. *Love, Sex and Choices* retained messaging about the risks of multiple partnership, but placed this within the narrative of a higher power sex script, emphasising sexual autonomy and sex-positive mechanisms through which prevention can be achieved [86–89]. *MyPEEPS mobile* achieved this integration of risk and pleasure through encouraging participants to think about their risk reduction and commit to how much sexual risk they are realistically willing to undertake [69–74].

Finally, interventions combined affirmation of capacity to engage with prevention with affirmation of marginalised identity. *Text Me Girl!*, an intervention targeted at young Trans women, integrated identity-affirming messages such as *'HIV meds can keep your Trans body strong and healthy'* and *'stay on top of your numbers with your doctor's help, now that's Trans Pride'* [90].

**3.2.2. Enjoyment.** 14 interventions operationalised enjoyment. These interventions focused on young people (n = 7; 50%), MSM, cisgender women (n = 4; 28.6%) and Trans women (n = 2; 14.3%), based in USA (n = 9; 64.3%), Ethiopia, Scotland, Zambia, Sweden and Australia (n = 1; 7.1%).

Interventions both focused on the role of desire in sexual decision-making and the role of prevention in enhancing enjoyment. *#PrEP4Love* affirms the role of physical pleasure in

sexual decision making, re-appropriating language common to risk-focused prevention programming and applying it to sex-positive messaging (i.e., *transmit love*) [48–50,91]. *Guy2Guy* affirms that we have sex because '*it feels good*', while also eroticising abstinence from penetrative sex, highlighting the pleasure associated with '*non-sex things that feel good sexually, like kissing and hand jobs*' [92–94]. Eroticising prevention is a key feature of other interventions, such as *SMS-Based Interventions on VMMC Uptake*, with messages such as '*Top reason is disease prevention, 2nd reason is sexual satisfaction*' [95]. Both interventions demonstrate the potential to promote risk-focused programming in a sex-positive manner.

Interventions also pre-empt and counter suggestions that condoms compromise pleasure. The Swedish application, *Skyddslaget*, devote a section to questioning norms and assumptions, specifically the assumption that sex with a condom is not good [76], and *Texting 4 Sexual Health* uses the message '*think using a condom will kill the mood? Getting an STI will kill the mood too. It's easier to enjoy sex when it's safe*' [77]. Some interventions also directly compare HIV prevention strategies with regard to enjoyment, with *Trans Women Connected* promoting the use of PrEP as an intervention that provides '*more intimacy*' and is '*better than condoms*' [96].

Interventions also provide details on how to enhance enjoyment when engaging with prevention, with *Guy2Guy* promoting the use of lubricant to enhance pleasure when using condoms [92–94].

Some interventions concurrently operationalise humour and enjoyment. *Make Your Position Clear* plays with the concept of positionality in sex between MSM with humorous taglines such as, '*Position #21, the watercooler*', drawing in elements of desire, as well as promoting assertiveness in the campaign title [97]. *Drama Downunder* also uses comedy, with one poster depicting a naked cisgender man with the lower half of his body reversed, with the title '*Get up-front about sexual health! You can get an STI in your arse even if you never get fucked!*' [98–100].

Seduction was a key tool in interventions, placing the target population as both the subject of seduction and empowering their capacity to seduce. *Text Me Girl!* incorporates this into its main focus of empowering Trans women, with one message empowering Trans women to '*be smart and sexy*' [90]. Contrastingly, *Ygetit?,* a mobile application based in the USA, promotes a digital comic strip, '*Tested*', which places the target population at the subject of seduction, showcasing a cisgender woman seductively encouraging a cisgender man she is buying condoms off of to accompany her to a safe sex van [101].

Enjoyment is finally operationalised in the platforms through which campaigns are advertised. *Make Your Position Clear* share their posters in saunas, bars and clubs; venues largely based around a pursual of physical pleasure [97].

**3.2.3. Emotional connection.** 9 interventions operationalised emotional connection. These interventions focused on MSM (n = 5; 55.6%), young people (n = 3; 33.3%) and Trans women (n = 2; 22.2%), with interventions based in USA (n = 5; 55.6%), China (n = 3; 33.3%) and Italy (n = 1; 11.1%, respectively).

A key feature was the promotion of protective mechanisms to enhance love between partners. The Chinese *Social Media Intervention Promoting HIV Testing* was based around the message that '*early prevention and early treatment will lead to long-lasting romance and long-lasting life*' [102]. Interventions also highlight the enhanced intimacy achieved through engaging with prevention alongside a partner. One Chinese HIV test promotion video targeting MSM shares the story of two individuals meeting and falling in love, and prior to engaging in a sexual relationship, they explore prevention tools and attend an HIV testing facility together, with the final message, '*being responsible for you makes our love lasting*' [103].

Interventions also emphasise the role prevention plays in protecting partners. *'Get an early check – chrysanthemum tea'* developed a 1-minute video presenting two MSM in a romantic relationship, promoting early detection as a way to prevent transmission to your partner [104]. *Text Me, Girl!* similarly promotes that *'when you stay in HIV care you can expose your heart, not your partner'* [90].

Finally, interventions detail how prevention can support greater trust between partners. For example, the *'Get an early check – chrysanthemum tea'* campaign suggests HIV testing will increase dyadic trust [104]. *mHealth-based approach as an HIV prevention strategy among people who use drugs on PrEP* also promotes the benefits of condoms in increasing comfort between partners, with one message reading *'condoms = peace of mind'* [105].

### 3.3. Efficacy of pleasure-based HIV PCCs

**3.3.1. Attitudinal change.** 14 papers investigated an outcome relevant to attitudinal change, including stigma, perceived norms, knowledge, self-efficacy and confidence.

Interventions reported positive effects on stigma, with statistically significant reductions in general stigma in *MyPEEPS Mobile* [71] and *Nalamanda Radio and Theatre Programme* [83]. *#PrEP4Love* also reported that young MSM who had seen their campaign were significantly more likely to perceive their gay and bisexual male (GBM) friends as strongly approving of PrEP, and more likely to perceive that their GBM friends were on PrEP [48].

The relationship between intervention exposure and perceived norms was inconsistent. Despite qualitative evidence suggesting that interventions such as *MyPEEPS Mobile* prompted participants to be *'a lot more cautious'* after the intervention, identifying that before intervention exposure they *'wouldn't use a condom because it wasn't shown a lot in [porn] and stuff'* [74], *Testing Makes Us Stronger* identified that among the same target population (MSM), there was no significant change in perceived norms surrounding condom use [80]. Similarly, although *Testing Makes Us Stronger* identified statistically significantly improved HIV testing norms [80], *People Like Us* found no significant difference in HIV testing norms between intervention and control [106]. Two aggregate measures of perceived norms measured in *Love, Sex and Choices* and *Guy2Guy*, measuring high-risk sex scripts and motivation to engage with prevention, respectively, found statistically significant improvements associated with intervention exposure [86,92].

Associations between HIV knowledge outcomes and intervention exposure were inconsistent. Qualitative accounts of *MyPEEPS Mobile* suggest intervention engagement was associated with increased knowledge about safe sex, HIV testing, biomedical prevention and the use of lubricants [74]. Similarly, those individuals with AIDS exposed to the *Nalamanda Radio and Theatre Programme* reported statistically significantly increased knowledge relating to HIV and ART [83]. However, although exposure to *Testing Makes Us Stronger* was statistically significantly associated with the belief that getting an HIV test was free and confidential, associations with other knowledge-related variables were insignificant [80]. Similarly, although participants exposed to *Trans Women Connected* had a significantly greater knowledge of PrEP, associations with other knowledge variables were insignificant [96], as were associations with exposure to *People Like Us* and knowledge-related variables concerning PrEP [106]. Finally, associations between exposure to *Texting for Sexual Health* and all knowledge-related variables concerning condom use were insignificant [77].

Similar inconsistencies were seen for self-efficacy. Exposure to *Testing Makes Us Stronger* was associated with statistically significant improvements in testing self-efficacy [80], however, this was not observed in *People Like Us* [106]. Exposure to *Safetxt* was associated with

significantly increased condom self-efficacy [66], but significance was not reached with exposure to *Just/Us* [84], *Texting for Sexual Health* [77] or *Safetxt* [66].

**3.3.2. Condom use.** 13 papers investigated an outcome related to condom use, including condom use at last sex, number of sex acts using a condom, number of condomless anal sex partners, condom use with new partners and condomless sex acts with partners of unknown HIV status.

Both interventions that investigated the association between intervention exposure and condom use at last sex, *Just/Us* and *Safetxt*, found statistically significant positive associations [66,84].

Associations between intervention exposure and number of condomless sex acts were inconsistent. Exposure to *Just/Us* and *Texting 4 Sexual Health* was associated with significantly increased odds of reporting using condoms with partners during follow-up [77,84]. Some interventions were efficacious in promoting condom use among only certain groups, such as *United Against AIDS* which was efficacious among the general population but not MSM or migrant communities [78]. Similarly, exposure to *Safer Sex Maintenance Text Messages* was significantly associated with reduced reporting of condomless sex among female sex workers in the Tijuana region of Mexico, but not in the Ciudad Juarez region [107]. Interestingly, exposure to *MyPEEPS Mobile* was initially associated with a statistically significant decrease in the number of condomless sex acts at three-months, but this significance was not observed at six- and nine-month follow-up. There were several interventions whose exposure was not significantly associated with change in condomless sex, including *Guide Enhanced Love, Sex and Choices* [87], *Nalamanda's Radio and Theatre Programme* [83], *People Like Us* [106], *InThisTogether* [108] and *Guy2Guy* [94].

The two studies quantitatively investigating the association between intervention exposure and condomless sex with partners of unknown HIV status found that exposure was insignificantly associated with having condomless sex with someone of unknown HIV status [78,87]. One qualitative response from a participant of *MyPEEPS Mobile*, however, articulated that following the intervention, they changed their practice, and, on one occasion, stopped having sex with someone because they *'didn't have a condom, [they] didn't know his status and [they] didn't] know [their own] either'* [71].

Both *MyPEEPS Mobile* [69] and *Crowdsourced HIV Test Promotion Video* [109] found a statistically insignificant association between exposure and the number of condomless anal sex partners.

Exposure to *Safetxt* was associated with significantly greater odds of reporting condom use at first sex with their most recent new partner [66].

**3.3.3. HIV testing.** 13 studies investigated an outcome related to HIV testing, investigating self-reported/clinic or city-level data on both self-administered and facility-based HIV testing.

Exposure to campaigns targeted at MSM was associated with significantly increased odds of self-reporting HIV testing in *Testing Makes Us Stronger* [81], *Social Media Intervention Promoting HIV Testing* [102], *Make Your Position Clear* [97], *Get an early check – chrysanthemum tea* [104], *Drama Downunder* [98,100] and *Guy2Guy* [94]. *Testing Makes Us Stronger* also demonstrated a significant increase in the odds of self-reporting test uptake between two- and six-month follow-up [81]. A statistically significant increase in the reporting of HIV testing was also observed among the general population in *United Against AIDS* (although this relationship was not significant on sub-group analyses of MSM or migrant communities) [78] and among young people exposed to *InThisTogether* [108].

*MyPEEPS Mobile* [69] and *Guide-Enhanced Love Sex and Choices* [87] found statistically insignificant associations between intervention exposure and HIV testing. Despite no

significant relationship between exposure to *People Like* Us and having *ever* tested for HIV, there was a statistically significant association with both testing regularly for HIV and reporting an intention to test for HIV [106].

Qualitative accounts of the effect of intervention exposure, specifically *MyPEEPS Mobile*, on HIV testing identified the role of the intervention in recognising the importance of testing, with one participant reporting that '*[MyPEEPS Mobile] gave me more of a priority to get tested to make sure that I know what's going on with my body, and making sure that I'm taking the necessary steps to stay this way*' [74].

**3.3.4. Health behaviour change.** 12 studies investigated an outcome related to health behaviour change, including changing interactions with healthcare providers, use of substances during sex, using measures to enhance the efficacy of prevention, VMMC, engagement with behavioural prevention campaigns and abstinence.

Three interventions reported participants changing behaviour towards healthcare providers after campaign exposure. *#PrEP4Love* exposure was associated with an increased likelihood of having conversations with medical providers about PrEP and an increased likelihood of being out to providers [48], *Nalamanda's Radio and Theatre Programme* was associated with significantly increased odds of asking doctors questions about HIV [83] and *Text Me Girl!* was associated with increased odds of attending an HIV care visit [90].

Four studies investigated the association between campaign exposure and the use of substances during sex. Despite qualitative accounts of *MyPEEPS Mobile* indicating the role of the intervention in helping individuals understand risks of sexualised drug use [71], quantitative investigations of the association between exposure to *Just/Us* [84], *MyPEEPS Mobile* [69] and *Guide-Enhanced Love, Sex and Choices* [87] and practicing sex under the influence of drugs or alcohol were all statistically insignificant.

*Safetxt* found no statistically significant association between exposure to the intervention and the use of lubricant for anal sex [97].

There was no statistically significant association between exposure to the *SMS-Based Interventions on VMMC Uptake* and the uptake of VMMC [95].

One study found a statistically significant association between intervention exposure and engagement with behavioural prevention campaigns, specifically finding a statistically significant increase in the number of calls to the Italian National AIDS Help-Line during the *United Against AIDS* campaign [78].

Both *InThisTogether* and *Guy2Guy* investigated the association between intervention exposure and abstinence, finding the association statistically insignificant [94,108]. Sexually experienced participants exposed to *Guy2Guy* were significantly *less* likely to report abstinence at follow-up [94].

**3.3.5. Biomedical prevention.** 6 studies investigated an outcome related to biomedical prevention, including the uptake of and adherence to both PrEP and ART.

There was discrepancy across studies investigating the association between campaign exposure and PrEP uptake. Although one analysis of *#PrEP4Love* found individuals exposed to campaigns were significantly more likely to initiate PrEP during follow-up [48], another study identified that the proportion of participants who saw *#PrEP4Love* adverts and subsequently initiated PrEP ranged from only 11.5–30.8% [91]. Qualitative accounts of *MyPEEPS Mobile* found participants felt the intervention would increase their likelihood to initiate PrEP [74], however, quantitative analyses identified a statistically insignificant association between *MyPEEPS Mobile* exposure and PrEP uptake [69].

There was an insignificant association between exposure to *Text Me, Girl!* and ART uptake and adherence [90].

**3.3.6. Partnership behaviour.** 4 studies investigated outcomes related to partnership, including change in the number of partners and number of new partners.

There was no clear association between intervention exposure and partnership behaviour. *Safetxt* and *Guide-enhanced Love, Sex and Choices* reported statistically significant decreases in partner numbers with intervention exposure [66,87]. However, *Just/Us* and *MyPEEPS Mobile* reported no statistically significant time*condition interactions [69,84]. In fact, a greater proportion of those exposed to *Just/Us* had had two or more sexual partners over 2-month and 6-month follow-up compared to controls, although this was statistically insignificant [84].

Exposure to *Safetxt* was associated with statistically significantly greater odds of not reporting sex with someone new during follow-up [66].

## 4. Discussion

This manuscript presents two narrative syntheses describing pleasure operationalisation within, and the efficacy of, pleasure-based HIV PCCs. Our findings have important implications for pleasure operationalisation. First, our inductive coding found interventions engaged with pleasure in a way that aligns with three key areas of existing definitions. The operationalisation of enjoyment affirmed the role of desire in sexual decision-making and promoted the role of prevention in enhancing pleasure, using humorous, seductive, and sex-positive messaging. Emotional connection focused on prevention's role in enhancing love and trust between partners, communicated through the role of prevention in protecting loved ones and through promoting the opportunity to engage with prevention alongside a partner. Empowerment affirmed sexual rights that need to be fulfilled for the realisation of pleasure, including autonomy, equality and self-determination, integrating self-efficacy training and harm reduction principles to achieve this. With many interventions deriving their operationalisations from focus groups and pilot testing, this indicates support from target communities for these components of pleasure. Whilst more research on the causal pathways linking pleasure and intervention efficacy is required, we propose that shifting focus to positive aspects of prevention, as demonstrated by included interventions, will better match people's lived realities of sexual experience, empower individuals to engage with care, give value to desire and affirm all consensual sexual encounters.

Regarding intervention efficacy, across most outcome measures, inconsistent associations were observed. There were a handful of outcome measures for which positive associations were observed for all investigated interventions, including condom use at last sex, positive interactions with healthcare providers and stigma. Given the small number of studies investigating each outcome, caution should be exercised when generalising these findings to all pleasure-based PCCs, and future work should further investigate these associations. Notably, some interventions observed differing impacts of intervention exposure on different subgroups. For example, exposure to *United Against AIDS* was associated with a statistically significant reduction in the number of condomless sex acts and increased HIV testing amongst the general population, but not amongst MSM or migrant communities [78]. Due to the intrinsically idiosyncratic nature of sexual pleasure [31], attitudes towards, and responses to, pleasure-based interventions are likely to vary across communities. Far from suggesting the restriction of pleasure-based interventions to specific groups, this finding should encourage increased investigation into differing responses to pleasure-based medicine within different communities. A dynamic, context-dependent and target-population-centred operationalisation of pleasure should be utilised when expanding interventions, developed in consultation with healthcare providers and target communities. Finally, there were important measures across which no statistically significant improvements were observed:

- **Sexualised drug and substance use**. Sexualised drug use has been labelled as an emerging public health crisis [110] due in part to its association with reduced condom use [111] and the transmission of HIV and other STIs [112]. Previous literature has noted that current public health initiatives insufficiently address sexualised drug use [113], reflected in the pleasure-based HIV PCCs included in this analysis.

- **ART uptake and adherence**. ART is a cost- and clinically-effective mechanism for the reduction of HIV transmission and for the management of AIDS-related complications [114], and pleasure-based interventions should target people living with AIDS and collect outcome data on secondary and tertiary prevention initiatives, such as the included *Nala-manda Theatre and Radio Programme* [83].

- **VMMC**. The efficacy of VMMC as an HIV prevention tool is contested [115], and the global VMMC campaign has been criticised for perpetuating a colonial-era power imbalance fuelled by the Global North and for interfering with the social landscape of communities in which circumcision holds cultural significance [116]. Given this, VMMC campaigns should not be promoted over other efficacious behavioural and biomedical interventions that are not associated with the same social, medical and political controversies.

There was notable disproportionate geographic representation among included studies, with 51.7% of interventions conducted in the USA. This proportion is, however, less than other systematic reviews on pleasure-based medicine, ranging from 75% [14] to 90% [15]. This is potentially due to our expansive definition of sexual pleasure which captured interventions focusing on sexual rights and empowerment. Despite this, only 6.9% of interventions were conducted in low-income countries (as determined by the World Bank Classification [117]). Although this is more than previous pleasure-based reviews [14,15], it suggests the extant research base is largely focused in higher-income settings [31]. Many cultures and settings have strong traditions of affirming sexual pleasure, and pleasure is an important determinant of sexual health, wellbeing and decision-making across a wide range of settings [118–120]. Included interventions demonstrating efficacy across low- and middle-income settings should inform future interventions, taking into consideration the sociocultural context they are being delivered in, and adapting intervention programming and pleasure operationalisation accordingly.

96.6% of interventions were targeted towards a specific population, with a particular focus on MSM, young people, people of colour and cisgender and Trans women. A focus on MSM, people of colour and young people matches findings from previous reviews on pleasure-based interventions [14], and historical trends in how these groups are over-sexualised [121]. This study highlighted more interventions targeted at cisgender and Trans women than previous reviews, potentially because of the focus on empowerment in our working definition of pleasure. There were notable demographic oversights in included interventions, however. Older individuals were not targeted in any interventions and were explicitly excluded in interventions such as *Ygetit?* [101] and *Texting 4 Sexual Health* [77]. Although sexual activity varies across the life-course, pleasure remains an important component of sexual well-being and decision-making well into later life [122]. The importance of pleasure in older communities, however, has long been neglected [14]. A presumed reduction in sexual activity among older communities can underly later detection and worse clinical outcomes of HIV for older people [123]. There were similar demographic oversights for women who have sex with women, Trans men and other gender divesre communities, and cisgender heterosexual men, all of whom have been identified as communities in which pleasure informs behaviour [31,124,125] and as important communities in HIV medicine [126–128]. Future interventions should

target these overlooked communities, as well as identifying common experiences of pleasure that can inform untargeted interventions, learning by example from untargeted interventions included in this review, such as *United Against AIDS* [78].

This review has identified the importance of not siloing HIV medicine from wider SRHR. This is of particular importance when promoting biomedical prevention, such as PrEP. Decades of *'AIDS exceptionalism'*, describing the phenomenon in which community perceptions of the risk of HIV/AIDS are elevated above other SRHR concerns [129], has led many to negate behavioural HIV prevention following the emergence of PrEP [130], due, in part, to community perceptions about the role of condoms in reducing sexual pleasure [4]. This review has identified interventions taking advantage of this. *Trans Women Connected* advocates PrEP's ability to *'increase intimacy'*, suggesting it's *'better than condoms'* [96]. This messaging undermines wider SRHR programming and increases the risk of STI transmission, which is of particular concern given the recent increasing incidence of anti-microbial resistant STIs, for which engagement with barrier methods are vital for prevention [131]. To achieve overall sexual well-being, it is vital that there is unification in HIV and SRHR preventive messaging. A delicate balance between promoting the potential increased pleasure associated with PrEP while continuing to encourage engagement with other relevant behavioural preventions must be struck, and those commissioning interventions should use examples of included interventions in this review that operationalise pleasure in promoting condom use, such as *Texting 4 Sexual Health* [77], to achieve this.

Finally, an upscaling of a pleasure-based approach to prevention must be complemented by a healthcare workforce that are prepared to discuss pleasure and practice in a sex-positive manner. Current medical training does not adequately incorporate discussions about pleasure or the importance of a sex-positivity [18,19,21]. A mismatch between the approaches of public health campaigns and the healthcare workforce risks reducing the efficacy of clinical encounters and undermining the messaging of both clinicians and public health campaigns. It is important that healthcare professionals engage with existing guidance on how to promote sex-positivity in their clinical encounters, such as the *Pleasure Project's* toolkit on *How to Become a Sexual Pleasure Champion and Trainer* [132].

There are important limitations to this study. It is possible that relevant articles were missed from our search strategy due to our single-screening approach, which increases risk of human error and bias [56]. However, our methodology aligns with recommendations to reduce the bias of single-screening, using a comprehensive secondary search strategy [56]. Our restriction to articles published only in peer-reviewed journals also excluded relevant interventions. For example, *PrEPverts*, a campaign that emphasises the connection between kink culture, passion, and PrEP use [133], has an operationalisation of pleasure with direct relevance to the study objectives. However, the decision to only include peer-reviewed was made to ensure the methodological rigour of included articles. Our date restriction excluded older pleasure-based HIV PCCs, such as the Terrence Higgins Trust's 1992 Campaign, *Safe Sex is Hot Sex* [134], but date restriction was necessary given the size of the extant literature. Relevant literature may have also been missed due to choices about study inclusion being influenced by the subjective nature of pleasure. This risk is reduced by our clear working definition of pleasure that guided study inclusion. No meta-analysis was performed on the outcome data extracted due to heterogeneity in the outcome measures. Inconsistent outcome reporting has been noted previously in the literature on pleasure-based medicine [14]. Although previous meta-analyses have been conducted in pleasure-based medicine, they have focused on a wider range of pleasure-based interventions investigating a narrower range of outcome measures, with both existing meta-analyses having only investigated condom use outcomes [14,15]. Thus, although the lack of a meta-analysis weakens the study, a relative strength of the study

is its focus on a wider range of reported outcomes. Finally, it is not possible to draw causal inferences on the direct effect of pleasure operationalisation from our findings, due to the lack of intervention studies with comparator groups exposed to an intervention that specifically does not operationalise pleasure, a limitation noted in previous reviews [14,21].

This study identifies three overarching operationalisations of pleasure in pleasure-based HIV PCCs; *Emotional Connection, Enjoyment* and *Empowerment*. All facets of pleasure demonstrated an integration of sex-positivity into campaigns, highlighting how prevention can both contribute to, and enhance, the sexual pleasure and sexual rights that individuals experience. This study promotes the importance of HIV PCCs, as well as wider aspects of HIV and SRHR, shifting focus to positive aspects of prevention that better match people's lived realities of sexual experience, empower individuals to engage with care and give value to desire, affirming all consensual sexual encounters. We have also identified the importance of a pleasure operationalisation that is dynamic, context-dependent and target-community-centred, with adaptable emphasis on the importance of sexual health, rights and pleasure, dependent on the priorities of target communities and the social and political climate in which interventions are delivered.

## Supporting information

**S1 File. PRISMA Checklist.**
(DOCX)

**S1 Table. Complete Search Strategy.**
(DOCX)

**S2 Table. Quality Assessment.**
(DOCX)

**S3 Table. Key Characteristics of Included Studies.**
(DOCX)

**S4 Table. Key Outcomes Reported by Included Studies.**
(DOCX)

**S5 Table. Details of Studies Excluded at Full-text Review.**
(XLSX)

## Acknowledgements

We extend the greatest thanks to subject experts we contacted throughout the project for direction and support with identifying literature for the secondary search strategy, including Dr Jaime García Iglesias (University of Edinburgh), Dr Richard Vytniorgu (University of Hertfordshire) and Dr Maurice Nagington (University of Manchester). We are also grateful to Dr Dean Connolly (London School of Hygiene and Tropical Medicine) for reviewing the manuscript and providing feedback to inform the final draft.

## Author contributions

**Conceptualization:** Luke Muschialli, Robert Pralat.

**Data curation:** Luke Muschialli.

**Formal analysis:** Luke Muschialli.

**Investigation:** Luke Muschialli.

**Methodology:** Luke Muschialli.

**Project administration:** Luke Muschialli.

**Supervision:** Jessie V. Ford, Lianne Gonsalves, Robert Pralat.

**Writing – original draft:** Luke Muschialli.

**Writing – review & editing:** Luke Muschialli, Jessie V. Ford, Lianne Gonsalves, Robert Pralat.

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
