## [Decision Letter · Decision Letter 0]

17 Jan 2025

Prevent with Pleasure : A systematic review of HIV public communication campaigns incorporating a pleasure-based approach

PGPH-D-24-02658

Dear Mr Muschialli,

We are pleased to inform you that your manuscript 'Prevent with Pleasure : A systematic review of HIV public communication campaigns incorporating a pleasure-based approach' has been provisionally accepted for publication in PLOS Global Public Health.

Best regards,

Tsitsi B. Masvawure, Ph.D.

Academic Editor

Thank you for your comprehensive and well written review. As you can see from the reviewer's comments, your review addresses an important issue and is very well done. I have reviewed your paper thoroughly and find it to be suitable for publication.

Reviewer Comments (if any, and for reference):

Reviewer's Responses to Questions

**Comments to the Author**

1. Does this manuscript meet PLOS Global Public Health’s publication criteria ? Is the manuscript technically sound, and do the data support the conclusions? The manuscript must describe methodologically and ethically rigorous research with conclusions that are appropriately drawn based on the data presented.

Reviewer #1: Yes

2. Has the statistical analysis been performed appropriately and rigorously?

Reviewer #1: N/A

3. Have the authors made all data underlying the findings in their manuscript fully available (please refer to the Data Availability Statement at the start of the manuscript PDF file)?

Reviewer #1: Yes

4. Is the manuscript presented in an intelligible fashion and written in standard English?

Reviewer #1: Yes

5. Review Comments to the Author

Reviewer #1: Dear Authors,

I believe your paper is ready to be published.

This paper addresses an important and often overlooked aspect of HIV public communication campaigns: the role of pleasure. The systematic review is well-conceived and effectively executed.

The arguments are clear and well-structured, supported by strong evidence, making this work a compelling contribution to the field. Highlighting pleasure as an integral part of HIV interventions is both innovative and necessary, filling a critical gap in current research and policy. My main suggestion, for future research, would be to conduct a similar review with a focus on the Global South. The authors have appropriately acknowledged this and other limitations in their discussion.

That said, there are a few minor adjustments to consider. On page 7, the acronym "SHR" should be spelled out as "sexual health and rights" to ensure clarity for all readers. Additionally, there should be consistency in capitalizing "Global North" throughout the paper.

This paper makes a significant contribution to research on HIV interventions by addressing a neglected yet essential topic. With these minor adjustments, it is ready for publication. I look forward to seeing it in print.

6. PLOS authors have the option to publish the peer review history of their article (what does this mean? ). If published, this will include your full peer review and any attached files.

**Do you want your identity to be public for this peer review?** For information about this choice, including consent withdrawal, please see our Privacy Policy .

Reviewer #1: **Yes: ** H Camilo Ruiz S
